# The impact of COVID-19 pandemic on key performance indicators in three Saudi hospitals

**Abeer Alharbi**[1]*, **Ranya S. Almana**[2], **Mohammed Aljuaid**[1]

**1** Department of Health Administration, College of Business Administration, King Saud University, Riyadh, Saudi Arabia, **2** MHA, King Fahad Medical City, Riyadh, Saudi Arabia

* aalharbi15@ksu.edu.sa

**Data Availability Statement:** All relevant data are within the paper and its Supporting information files.

**Funding:** Initials of the authors who received each award. AA, RM, MA Grant numbers awarded to

## Abstract

### Objective

The coronavirus disease (COVID-19) disrupted healthcare systems and medical care worldwide. This study attempts to assess the performance of three Saudi hospitals during COVID-19 by comparing waiting times for outpatient appointments and the volume of elective surgeries before and after COVID-19.

### Methods

We used ADA'A data collected from three Saudi hospitals for this retrospective cohort study. The outcome variables were "Waiting Time for Appointment" and "Elective OR Utilization". The hospitals included in this study were: a 300-bed maternity and children's hospital; a 643-bed general hospital; and a 1230-bed tertiary hospital. We included all patients who visited the OPD and OR in the time period from September 2019 to December 2021. A two-way ANOVA test was used to examine the differences in the outcome variables by hospital and by the phase of COVID-19.

### Results

For the elective OR utilization rate, the results showed that both the hospital and the phase of COVID-19 were significantly different (p-value < 0.05). On average, the elective OR utilization rate dipped considerably in the early phase of COVID-19 (33.2% vs 44.9%) and jumped sharply in the later phase (50.3%). The results showed that the waiting time for OPD appointment was significantly different across hospitals and before and after COVID-19 in each hospital (p-value < 0.05). the waiting time dropped during the early phase of COVID-19 for both the general hospital (GEN) (24.6 days vs 34.8 days) and the tertiary hospital (MDC) (40.3 days vs 48.6 days), while the maternity and children's hospital (MCH)'s score deteriorated sharply (24.6 days vs 9.5 days).

### Conclusion

This study indicates that COVID-19 led to a significant impact on elective surgery rates and waiting time for OPD appointments in the early stage of the pandemic when the lockdown

each author. Funded through the Researcher Supporting Project (RSP2022R481), King Saud University, Riyadh, Saudi Arabia. The full name of each funder. Funded through the Researcher Supporting Project (RSP2022R481), King Saud University, Riyadh, Saudi Arabia. URL of each funder website. https://dsrs.ksu.edu.sa/en Did the sponsors or funders play any role in the study design, data collection and analysis, decision to publish, or preparation of the manuscript? NO - The funders had no role in study design, data collection and analysis, decision to publish, or preparation of the manuscript.

**Competing interests:** The authors have declared that no competing interests exist.

strategy was implemented in the country. Although the elective surgery rate had decreased at the designated COVID-hospital, the waiting time for OPD appointment had improved. This is a clear indication that the careful planning and management of resources for essential services during pandemic was effective.

## Introduction

The coronavirus disease (COVID-19) disrupted healthcare systems and medical care worldwide. The virus was first reported in Wuhan, China, on December 13, 2019. In Saudi Arabia, the first case was confirmed in March 2020 with the number of such cases reaching its peak in June 2020 and fluctuating during July of that year [1]. Then, from August 2020, the number of confirmed cases declined and reached the minimum in February 2021 [1]. From Mar 2 to July 25 in 2020, there have been 262,772 confirmed cases of COVID-19 with 2,672 deaths [2]. In March 8, 2020, the Saudi authorities adopted the lockdown strategy to contain the spread of the disease. They also introduced a range of public health measures consisting of complete and partial curfews, the closure of educational institutions, social distancing, the mandatory use of masks, and the suspension of mass gatherings (like praying in mosques), domestic and international flights, and social and sports events. Hospitals in Saudi Arabia recommended minimizing nonessential visits, especially for children and the more vulnerable persons, to decrease the spread of the virus and to ensure that there was enough capacity to handle the surges in COVID-19 cases. To maximize patient and health worker safety, modifications to service delivery included identifying nonessential health services that could be delayed or canceled in accordance with local or national guidance. Postponing nonessential health services freed health workers to provide COVID-19 care and reduced crowding in waiting rooms. As a result, routine outpatient visits and elective surgeries were postponed [3]. Waiting times for outpatient appointments and the volume of elective surgeries performed were strongly affected by the pandemic [4–6].

This study attempts to assess the performance of three Saudi hospitals during COVID-19 by comparing waiting times for outpatient appointments and the volume of elective surgeries before and after COVID-19. This was done with a view of helping policymakers to better understand the impact of the pandemic on the healthcare system so they can prepare contingency plans for any such future pandemics. In addition, the inclusion in the study of three hospitals with different scopes of services provided an opportunity to assess the performance in different healthcare settings.

## Materials and methods

### Data source

We used ADA'A data collected from three Saudi hospitals for this retrospective cohort study. The Ministry of Health (MOH) in Saudi Arabia launched the ADA'A (*Performance*) program in 2015 to improve the operational performance of the MOH hospitals in Saudi Arabia. The ADA'A program collects several Key Performance Indicators (KPIs) as a means of evaluating the progress of the health system. The KPIs were developed for use by the hospitals to monitor, evaluate, and improve their performing against established benchmarks.

## Outcome variables

The KPIs "Waiting Time for Appointment" and "Elective OR Utilization" reflect the performance of the outpatient department (OPD) and the operating room (OR), respectively. "Waiting Time for Appointment" estimates the waiting time (in days) for OPD appointment by specialty, as defined by the third available appointment in OPD per specialty. ADA'A used these following four benchmarks to evaluate this KPI: world Class (less than 14 days), acceptable (14–28 days), need improvement (28–42 days), and unacceptable (more than 42 days). "Elective OR Utilization" estimates the percentage of hours the operation rooms were occupied for elective surgeries within standard hours. The following four benchmarks were used to evaluate this KPI: world class (more than 75%), acceptable (62.5% to 75%), need improvement (50.1% to 62.5%), and unacceptable (<50%). Tables 1 and 2 provide details on the sample size required for the data to be collected, the inclusion and exclusion criteria, and the benchmark classification and target.

## Setting

The hospitals included in this study were: a 300-bed maternity and children's hospital (MCH); a 643-bed general hospital (GEN); and a 1230-bed tertiary hospital (MDC). We used data from all patients who visited the OPD and OR in the time period from September 2019 to December 2021.

## Statistical analysis

The data were analyzed using the Statistical Package for Social Sciences (SPSS) program, version 25. A two-way ANOVA test was used to examine the differences in the outcome variables by hospital and by the phase of COVID-19 which was divided into three phases: Pre- COVID (9/2019–2/2020) coded as 0; early-COVID (3/2020–7/2020) coded as 1; and later-COVID (8/2020–12/2021) as 2. A p-value of less than 0.05 was considered to be statistically significant. Effects plots with fitted means and confidence intervals (CI) were used to explain the difference in the outcome variable by hospital and by the phase of COVID-19.

## Ethical considerations

The study was reviewed and approved by the Ethics Committee of Scientific Research that supports the King Fahad Medical City Institutional Review Board (KFMD IRB). The committee, on behalf of the Institutional Review Board, approved the research (reference number 22-

**Table 1. Data collection/measurement in OPD.**

| KPI 1: Waiting Time for Appointment (3rd Available Appointment) | |
|---|---|
| Sample size | 100% |
| Calculation | Waiting time (in days) for OPD appointment by specialty, as defined by the 3rd available appointment in ODP per specialty. This KPI is measured once per week; the "3rd next available appointment" date is recorded for every specialty. This must be measured on the same day of the week each week, in order to ensure consistency of measurement. |
| Exclusion | Anesthesia clinic (due to scheduling appointments for one week before surgery) |
| World class | Less than 14 days |
| Acceptable | 14–28 days |
| Need Improvement | 28–42 days |
| Unacceptable | More than 42 days |

**Table 2. Data collection/measurement in OR.**

| KPI 2: Elective OR Utilization | |
|---|---|
| Sample size | 100% |
| Numerator | the sum of hours rooms occupied "wheels in" to "wheels out" within standard working hours |
| Denominator | Sum of hours available (total number of functional rooms* 8 working hours a day* number of working days a week) |
| Calculation | [Numerator/ denominator] × 100 |
| Inclusion | Elective surgeries performed within standard hours. Also, surgeries that start within working hours, but overrun |
| Exclusion | All emergency surgeries |
| World class | More than 75% |
| Acceptable | 62.5% to 75% |
| Need Improvement | 50.1% to 62.5% |
| Unacceptable | <50% |

347E). The need for consent was waived by the ethics committee as we are reporting a retrospective study of quality metrics reports where all data were fully anonymized before accessing them.

## Results

The data were collected over 28 months beginning in September of 2019. The elective OR utilization rate and the waiting time for OPD appointment (in days) were plotted by hospital across the 28-month period [Figs 1 and 2]. To examine the differences in the outcome variables by hospital and by the phase of COVID-19, a two-way ANOVA test was used.

### Elective OR utilization rate

For the elective OR utilization rate, the results showed that both the hospital and the phase of COVID-19 were significantly different (p-value < 0.05) [Table 3]. The effects plots tell how they were different [Fig 3]. On average, the elective OR utilization rate dipped considerably in the early phase of COVID-19 (33.2% vs 44.9%) and jumped sharply in the later phase (50.3%). Additionally, the average scores for the maternity and children's hospital (MCH) (25.5%) were sharply lower than both the general hospital (GEN) and the tertiary hospital (MDC) (55.3%, and 47.5%, respectively). Looking at the interaction plots, the utilization rate dropped during the early phase of COVID-19 for both general hospital (GEN) (44.4% vs 66.6%) and tertiary hospital (MDC) (31.5% vs 49.6%), while the maternity and children's hospital (MCH) percentages rose slightly (23.7% vs 18.4%). Then in the later phase, GEN, MDC, and MCH all improved their level (54.9%, 61.5, and 34.4%, respectively).

### Waiting time for OPD appointment

The results showed that the waiting time for OPD appointment was significantly different across hospitals and before and after COVID-19 in each hospital (p-value < 0.05) [Table 3]. The effects plots tell how they were different [Fig 4]. The average scores for the maternity and children's hospital (MCH) (16.7 days) were sharply lower than both the general hospital (GEN) and the tertiary hospital (MDC) (32.9 days, and 36.8 days, respectively). Looking at the interaction plots, the waiting time dropped during the early phase of COVID-19 for both the general hospital (GEN) (24.6 days vs 34.8 days) and the tertiary hospital (MDC) (40.3 days vs 48.6 days), while the maternity and children's hospital (MCH)'s score deteriorated sharply

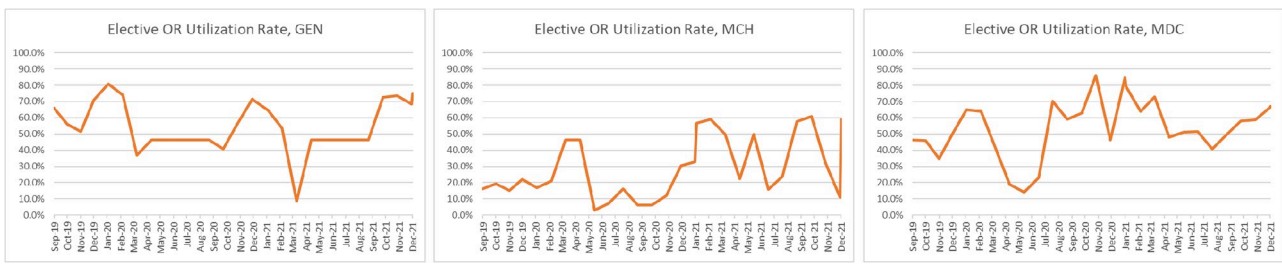

**Fig 1. The elective OR utilization rate plotted by hospital across the 28-month period.**

(24.6 days vs 9.5 days). Then in the later phase, MDC's score continued to improve by falling further to 21.4 days. Additionally, MCH's score improved by being lowered to 15.9 days while GEN score deteriorated as it increased to 39.2 days.

## Discussion

For elective surgery rate, the study findings revealed that there was a variation before and after COVID-19. Overall, there was a significant decrease in the elective surgery rate during the early stage of COVID-19. The lower rate of elective surgeries was probably caused by the changes implemented at the Saudi hospitals including postponing elective surgeries in response to the spread of the disease. This is consistent with previous studies which discovered that the rate of elective surgeries was negatively affected by COVID-19 [4, 5]. The study findings also revealed that the performance seemed to vary significantly across hospitals. This was not surprising considering the dissimilarity in the scope of services provided in each of them, the population served, and the complexity of the medical condition. In addition, the general hospital (GEN) was assigned as a COVID-hospital while the other two were not. Looking at the effect of COVID-19 on each hospital, we observed that during the early stage of COVID-19, both the general hospital (GEN) and the tertiary hospital (MDC)'s elective surgery rate had deteriorated. The rate dropped sharply for GEN going from "acceptable" at pre-COVID-19 to "unacceptable" in early COVID-19 stage. In addition, MDC performance dropped by eighteen percentage points in early COVID-19. These drops in elective surgery rates were to be expected as healthcare systems were affected by the pandemic with the surges of COVID-19 cases and the newly implemented guidelines and mitigation precautions implemented by the OR. These changes included decreasing the hospitals utilization to 50%, and in GEN hospital, the majority of the manpower was transferred to critical care units because it was designated as a COVID-19 hospital. In the later stage of COVID-19, however, both GEN and MDC improved their elective surgery rates with the post-lockdown abatement of COVID-19.

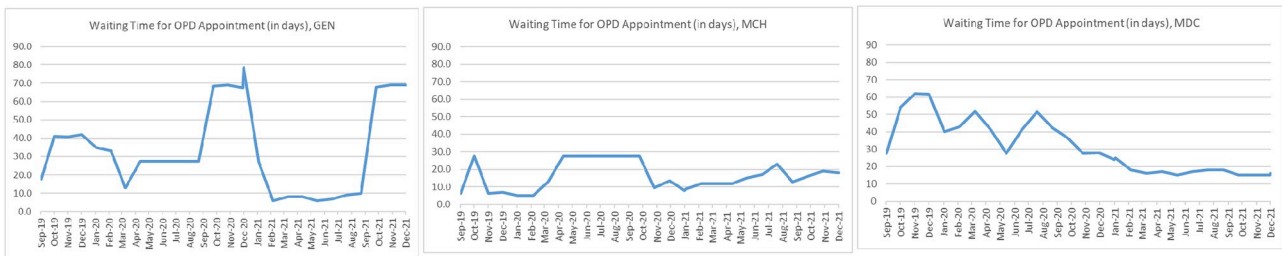

**Fig 2. The waiting time for OPD appointment (in days) plotted by hospital across the 28-month period.**

**Table 3. Analysis of variance (ANOVA).**

| Source | DF | SS | MS | F | P |
|---|---|---|---|---|---|
| **Elective OR Utilization Rate** | | | | | |
| Pre/Post-COVID-19 | 2 | 0.32 | 0.16 | 6.19 | 0.0033 |
| Hosp. Type | 2 | 1.01 | 0.50 | 19.31 | <0.001 |
| Interaction | 4 | 0.28 | 0.07 | 2.66 | 0.0391 |
| Error | 75 | 1.95 | 0.03 | | |
| Total | 83 | 3.78 | 0.05 | | |
| **Waiting time for OPD appointment** | | | | | |
| Pre/Post-COVID-19 | 2 | 509.15 | 254.58 | 1.01 | 0.3677 |
| Hosp. Type | 2 | 4742.2 | 2371.1 | 9.45 | <0.001 |
| Interaction | 4 | 4960.3 | 1240.1 | 4.94 | 0.0014 |
| Error | 75 | 18829 | 251.06 | | |
| Total | 83 | 30230 | 364.22 | | |

Nevertheless, MCH's performance improved in the early stages of COVID-19 as the rate of elective surgeries increased by five percentage points. MCH is a 300-bed maternity and children's hospital that offers elective surgeries including: Lower Segment Caesarean Section (LSCS), Myomectomy, Dilation and Curettage (D&C), Polypectomy, and Suction Curettage. The increase in the rate of elective surgeries at MCH is consistent with another study that found caesarean sections and facility-based deliveries showed significant increases during COVID-19 [7]. In the later stage of COVID-19, the elective surgery rate continued to improve. Despite the increase in the elective surgery rate, MCH still scored a less than 50 percent utilization rate. The low acuity of patients presenting at the hospital might be the reason why less patients were having elective surgeries as compared to the other two hospitals which accommodated more complex patient conditions.

As for waiting time for OPD appointments, we observed that during the early stage of COVID-19, both general hospital (GEN) and tertiary hospital (MDC)'s performance had improved. The waiting time in days dropped sharply for GEN going from "need improvement" at pre-COVID-19 to "acceptable" in the early COVID-19 stage. In addition, MDC's waiting times dropped by eight days on average in the early COVID-19 stage, moving up to the "need improvement" category. These drops in waiting times for OPD appointments were to be expected as people were avoiding hospitals during the pandemic and appointments were cancelled for mild cases in response to the spread of the disease. This was particularly true for GEN hospital which became a designated COVID-hospital focusing on critical care only. The total number of OPD visits dropped sharply in the early stage of COVID-19 going from more

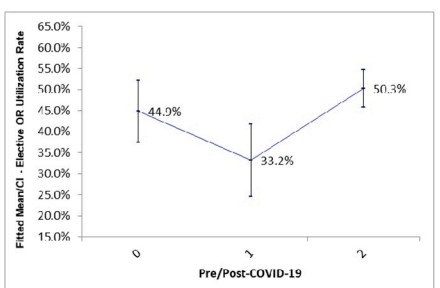 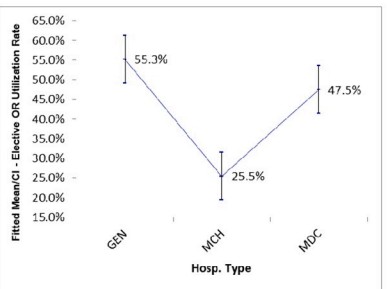 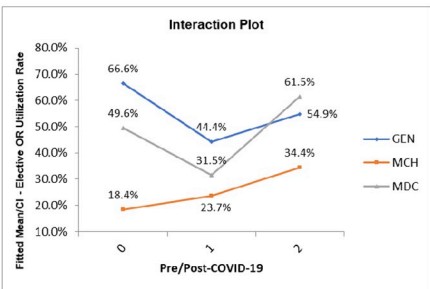

**Fig 3. Effects plots for elective OR utilization rate by hospital and phase of COVID-19.**

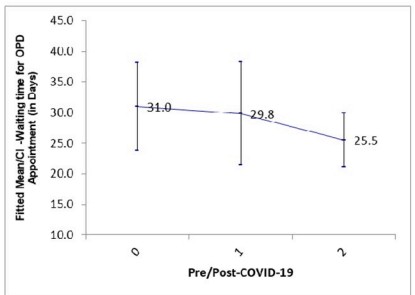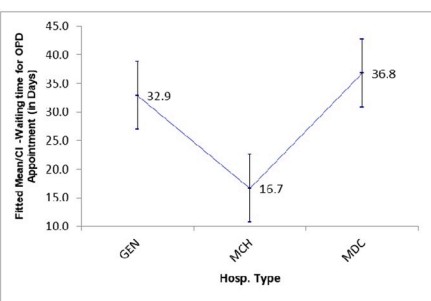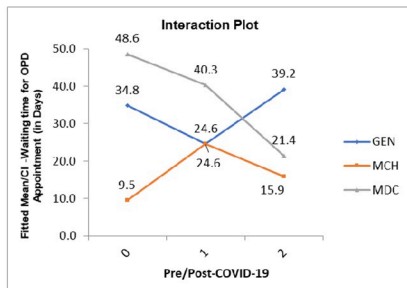

**Fig 4. Effects Plots for waiting time for OPD appointment (in days) by hospital and phase of COVID-19.**

than 89,000 visits to less than 14,000 [S1 Table]. In MDC, the OPD was shut down completely for 2 weeks from mid-March until end of April and they have initiated a virtual clinic for all specialties, except oncology and chemotherapy, which may explain the drop in OPD waiting times. Similarly, the total number of OPD visits dropped sharply in the early stage of COVID-19 going from around 272,000 visits to less than 26,000. However, MCH's waiting time in days had increased by 15 days on average. MCH restricted its outpatient appointments for only gynecology and part of these appointments were done through virtual visits. In the later stage of COVID-19, MCH improved its waiting time and GEN increased its average waiting time for OPD appointments with the post-lockdown abatement of COVID-19.

This study had some limitations in that it was conducted in the OR and OPD of a specific geographical area of Saudi Arabia which may affect the generalizability of the study. In addition, due to the study's retrospective nature, limited information was collected on some areas. We used a database of the quality metrics and not medical charts; therefore, the lack of individual patient characteristics prevented us from performing further analysis as to the possible reasons for the OR utilization rates and OPD waiting times in terms of diagnosis, acuity level, and demographics. Future studies taking these factors in consideration are recommended.

## Conclusions

The current study indicates that COVID-19 led to a significant impact on elective surgery rates and waiting time for OPD appointments in the early stage of the pandemic when the lockdown strategy was implemented in the country. The study indicates that the general and tertiary hospitals had a decreased elective surgery rate and waiting time for OPD appointments during the early stage of the pandemic, while the maternity and children's hospital had an increased elective surgery rate and waiting time for OPD appointments. Although the elective surgery rate had decreased at the designated COVID-hospital, the waiting time for OPD appointment had improved. This is a clear indication that the careful planning and management of resources for essential services during pandemic was effective. These findings have provided a better understanding of the impact of the pandemic on the healthcare system and should allow policymakers to prepare contingency plans for any such future pandemics. A well-planned contingency strategy that relies on the strength of the health system and the competence of health personnel is recommended. The plan should provide timely response that is accomplished through careful resource planning, coordination and monitoring at the national level, community involvement, and epidemiological surveillance. Rapid transformation and careful allocation of resources together with changing working methods in hospitals have the potential to mitigate challenges presented by the pandemic.

## Supporting information

**S1 Table. Total number of OPD and OR visits in each COVID-19 stage in each hospital.** (DOCX)

**S1 Data.** (XLSX)

## Author Contributions

**Conceptualization:** Abeer Alharbi, Ranya S. Almana, Mohammed Aljuaid.

**Data curation:** Ranya S. Almana.

**Formal analysis:** Abeer Alharbi, Ranya S. Almana, Mohammed Aljuaid.

**Methodology:** Abeer Alharbi, Ranya S. Almana, Mohammed Aljuaid.

**Writing – original draft:** Ranya S. Almana, Mohammed Aljuaid.

**Writing – review & editing:** Abeer Alharbi.

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
