## [Decision Letter · Decision Letter 0]

8 Jan 2023

PONE-D-22-26363The Impact of COVID-19 Pandemic on Key Performance Indicators in Three Saudi HospitalsPLOS ONE

Dear Dr. Alharbi,

Thank you for submitting your manuscript to PLOS ONE. After careful consideration, we feel that it has merit but does not fully meet PLOS ONE’s publication criteria as it currently stands. Therefore, we invite you to submit a revised version of the manuscript that addresses the points raised during the review process.

We look forward to receiving your revised manuscript.

Kind regards,

Nasser Hadal Alotaibi

Academic Editor

PLOS ONE

Journal Requirements:

Reviewers' comments:

Reviewer's Responses to Questions

**Comments to the Author**

1. Is the manuscript technically sound, and do the data support the conclusions?

Reviewer #1: Partly

Reviewer #2: Yes

2. Has the statistical analysis been performed appropriately and rigorously? 

Reviewer #1: Yes

Reviewer #2: Yes

3. Have the authors made all data underlying the findings in their manuscript fully available?

Reviewer #1: No

Reviewer #2: Yes

4. Is the manuscript presented in an intelligible fashion and written in standard English?

Reviewer #1: Yes

Reviewer #2: Yes

5. Review Comments to the Author

Reviewer #1: 1. Table 1 Data Collection, in the sample size category, the author wrote 100% without stating the number of sample size? Does the 100% is referring to the sample size obtained? Best if the author can state the number of sample obtained in frequency too. The sample size obtained for the study is unclear. Is the bed capacity in the respective hospitals is your actual sample?

2. Table 3 Analysis of Variance; best if the author can limit the decimal point to the two decimal point for standardization in statistic. Please follow journal requirement. If it is not stated, two decimal point is best used to indicate the statistic results.

3. Paragraph for discussion (line 185 - 193) is not needed. Focus your discussion based on the result without stating facts that have been previously discussed earlier in the introduction section.

4. The paper is lacking in critical recommendation on the issue, while the author is pushing for proposed contingency plan? what kind of contingency plan that should be proposed for the pandemic? This might add value to the policy maker. Thus the conclusion seemed lacking in providing concrete ending to what should be done to prevent/ address the issue.

Reviewer #2: A well organized article. The ethical review for the research meets the required standards. The methods of statistical analysis used for the research were sound and the results of the 2 way ANOVA test supported the conclusion of the article. Enough data was provided in the article although it would have been easier to absorb/understand the provided data used for analysis (OPD waiting times in days and Elective OR utilization rates during Pre, early and post pandemic periods) if they were all included in a single table of data extraction. The sizes of figures 1-4 could be increased a bit to provide better visibility for the reader. The article was written intelligibly, with appropriate use of the English language although there unnecessary repetition of facts in the introduction and discussion sections eg ''To contain the spread of the COVID-19 disease, the Saudi government adopted the lockdown strategy''. Adequate explanation of the reasons for variation in the results/findings among the 3 hospitals were give in the discussion section with good interpretation of the findings. The limitations were addressed properly and the conclusion was brief and concise. In general, it was a very interesting and quite unique article as it aims to explore, compare and analyse the differences in pandemic effects on Key performance indicators in 3 hospitals providing very different types of medical care.

6. PLOS authors have the option to publish the peer review history of their article (what does this mean?). If published, this will include your full peer review and any attached files.

Reviewer #1: **Yes: **Mohd Redhuan Dzulkipli

Reviewer #2: **Yes: **Atinuke Hephzibah Adeyemo

---

## [Author Response · Author response to Decision Letter 0]

16 Jan 2023

Response to Reviewers

Reviewer #1

Thank you so much for your valuable comments. Below are my responses for each point.

1. Table 1 Data Collection, in the sample size category, the author wrote 100% without stating the number of sample size? Does the 100% is referring to the sample size obtained? Best if the author can state the number of sample obtained in frequency too. The sample size obtained for the study is unclear. Is the bed capacity in the respective hospitals is your actual sample?

For our research, we used secondary data from Adaa’ program, collected to measure hospital quality metrics. This data did not report the exact sample size but did explain that for the OPD indicator, the data collectors used 100% of the patient’s data who used the OPD during the study period. Similarly, for the OR indicator, the data collectors used 100% of the patient’s data who used the OR during the study period. However, we added the visits numbers for the OPD and OR during the study three stages (pre-COVID, early-COVID, and post-COVID) in the supplement file appendix 1.

2. Table 3 Analysis of Variance; best if the author can limit the decimal point to the two decimal point for standardization in statistic. Please follow journal requirement. If it is not stated, two decimal point is best used to indicate the statistic results.

Corrected as advised.

3. Paragraph for discussion (line 185 - 193) is not needed. Focus your discussion based on the result without stating facts that have been previously discussed earlier in the introduction section.

Deleted as advised. 

4. The paper is lacking in critical recommendation on the issue, while the author is pushing for proposed contingency plan? what kind of contingency plan that should be proposed for the pandemic? This might add value to the policy maker. Thus the conclusion seemed lacking in providing concrete ending to what should be done to prevent/ address the issue.

Recommendations has been added to the conclusion section (Lines 254-260)

Reviewer #2

A well organized article. The ethical review for the research meets the required standards. The methods of statistical analysis used for the research were sound and the results of the 2 way ANOVA test supported the conclusion of the article. Enough data was provided in the article although it would have been easier to absorb/understand the provided data used for analysis (OPD waiting times in days and Elective OR utilization rates during Pre, early and post pandemic periods) if they were all included in a single table of data extraction. The sizes of figures 1-4 could be increased a bit to provide better visibility for the reader. The article was written intelligibly, with appropriate use of the English language although there unnecessary repetition of facts in the introduction and discussion sections eg ''To contain the spread of the COVID-19 disease, the Saudi government adopted the lockdown strategy''. Adequate explanation of the reasons for variation in the results/findings among the 3 hospitals were give in the discussion section with good interpretation of the findings. The limitations were addressed properly and the conclusion was brief and concise. In general, it was a very interesting and quite unique article as it aims to explore, compare and analyse the differences in pandemic effects on Key performance indicators in 3 hospitals providing very different types of medical care.

Thank you so much for your valuable comments. 

The data used for analysis (OPD waiting times in days and Elective OR utilization rates during Pre, early and post pandemic periods) were all included in a supplemental file. 

The figures have been increased in size for better visibility. 

Also, the unnecessary repetition of facts in the discussion section has been deleted.

---

## [Decision Letter · Decision Letter 1]

27 Apr 2023

The Impact of COVID-19 Pandemic on Key Performance Indicators in Three Saudi Hospitals

PONE-D-22-26363R1

Dear Dr. Alharbi,

We’re pleased to inform you that your manuscript has been judged scientifically suitable for publication and will be formally accepted for publication once it meets all outstanding technical requirements.

Kind regards,

Nasser Hadal Alotaibi

Academic Editor

PLOS ONE

Additional Editor Comments (optional):

Reviewers' comments:

Reviewer's Responses to Questions

**Comments to the Author**

1. If the authors have adequately addressed your comments raised in a previous round of review and you feel that this manuscript is now acceptable for publication, you may indicate that here to bypass the “Comments to the Author” section, enter your conflict of interest statement in the “Confidential to Editor” section, and submit your "Accept" recommendation.

Reviewer #1: All comments have been addressed

2. Is the manuscript technically sound, and do the data support the conclusions?

Reviewer #1: Yes

3. Has the statistical analysis been performed appropriately and rigorously? 

Reviewer #1: Yes

4. Have the authors made all data underlying the findings in their manuscript fully available?

Reviewer #1: Yes

5. Is the manuscript presented in an intelligible fashion and written in standard English?

Reviewer #1: Yes

6. Review Comments to the Author

Reviewer #1: (No Response)

7. PLOS authors have the option to publish the peer review history of their article (what does this mean?). If published, this will include your full peer review and any attached files.

Reviewer #1: No

---

## [Editor Report · Acceptance letter]

2 May 2023

PONE-D-22-26363R1 

The Impact of COVID-19 Pandemic on Key Performance Indicators in Three Saudi Hospitals 

Dear Dr. Alharbi:

I'm pleased to inform you that your manuscript has been deemed suitable for publication in PLOS ONE. Congratulations! Your manuscript is now with our production department. 

Kind regards, 

on behalf of

Dr. Nasser Hadal Alotaibi 

Academic Editor

PLOS ONE